# Cholinergic input to mouse visual cortex signals a movement state and acutely enhances layer 5 responsiveness

**Baba Yogesh[1,2], Georg B Keller[1,2]***

[1]Friedrich Miescher Institute for Biomedical Research, Basel, Switzerland; [2]Faculty of Natural Sciences, University of Basel, Basel, Switzerland

*For correspondence:
georg.keller@fmi.ch

Competing interest: The authors declare that no competing interests exist.

**Abstract** Acetylcholine is released in visual cortex by axonal projections from the basal forebrain. The signals conveyed by these projections and their computational significance are still unclear. Using two-photon calcium imaging in behaving mice, we show that basal forebrain cholinergic axons in the mouse visual cortex provide a binary locomotion state signal. In these axons, we found no evidence of responses to visual stimuli or visuomotor prediction errors. While optogenetic activation of cholinergic axons in visual cortex in isolation did not drive local neuronal activity, when paired with visuomotor stimuli, it resulted in layer-specific increases of neuronal activity. Responses in layer 5 neurons to both top-down and bottom-up inputs were increased in amplitude and decreased in latency, whereas those in layer 2/3 neurons remained unchanged. Using opto- and chemogenetic manipulations of cholinergic activity, we found acetylcholine to underlie the locomotion-associated decorrelation of activity between neurons in both layer 2/3 and layer 5. Our results suggest that acetylcholine augments the responsiveness of layer 5 neurons to inputs from outside of the local network, possibly enabling faster switching between internal representations during locomotion.

## eLife assessment

This **fundamental** study by Yogesh and Keller provides a set of results describing the response properties of cholinergic input and its functional impacts in the mouse visual cortex. They found that cholinergic inputs are elevated by locomotion in a binary manner regardless of locomotor speeds, and activation of cholinergic input differently modulated the activity of Later 2/3 and Layer 5 visual cortex neurons induced by bottom-up (visual stimuli) and top-down (visuomotor mismatch) inputs. The experiments are cutting-edge and well-executed, and the results are **convincing**.

## Introduction

Acetylcholine is one of the key neuromodulators involved in cortical function and plasticity. A rich body of work has shown that acetylcholine gates experience-dependent changes in the responses of cortical neurons to sensory inputs. This includes receptive field plasticity in auditory cortex (*Bakin et al., 1996*; *Froemke et al., 2007*; *Kilgard and Merzenich, 1998*; *Metherate and Weinberger, 1990*), somatosensory cortex (*Ego-Stengel et al., 2001*; *Rasmusson and Dykes, 1988*; *Sachdev et al., 1998*), and olfactory cortex (*Wilson et al., 2004*), or ocular dominance plasticity in visual cortex (*Bear and Singer, 1986*; *Greuel et al., 1988*; *Gu and Singer, 1993*; *Kirkwood et al., 1999*). In all cases, the central tenet is that pairing of acetylcholine release or its application concurrent with a sensory stimulus results in plasticity that increases the responsiveness of neurons to subsequent presentations of the same stimulus. While the evidence that acetylcholine gates plasticity is overwhelming and

unequivocal, it remains less well understood what drives the release of acetylcholine in cortex during behavior and what its functional consequences are on cortical computations acutely.

Cholinergic projections to cortex arise from a population of neurons in the basal forebrain (*Li et al., 2018*). Within this population of neurons, there is a relatively broad functional heterogeneity (*Laszlovszky et al., 2020*; *Robert et al., 2021*), with responses observed during movement – be it locomotion or licking (*Harrison et al., 2016*), to reinforcers (*Hangya et al., 2015*), to auditory stimuli (*Guo et al., 2019*; *Robert et al., 2021*), and weakly to visual stimuli (*Robert et al., 2021*). However, inferring acetylcholine release in specific cortical targets based on the activity of cholinergic neurons within the basal forebrain is complicated by the fact that different regions (*Kim et al., 2016*; *Pinto et al., 2013*) and cholinergic cell types (*Laszlovszky et al., 2020*) in the basal forebrain project to different areas of cortex. This projection specificity likely underlies the heterogeneity in cholinergic activity across dorsal cortex (*Collins et al., 2023*; *Lohani et al., 2022*). Measurements of cholinergic activity in cortex have found that movement in the form of a lever press is associated with cholinergic activity in mouse motor cortex, somatosensory cortex, and prefrontal cortex (*Ren et al., 2022*). Whisking is associated with cholinergic axonal activity in barrel cortex (*Eggermann et al., 2014*), while sound (*Zhu et al., 2023*) and movement (*Nelson and Mooney, 2016*; *Zhu et al., 2023*) are associated with cholinergic activity in auditory cortex, and locomotion (*Larsen et al., 2018*; *Lohani et al., 2022*), facial movement (*Lohani et al., 2022*), and pupil dilation (*Larsen et al., 2018*; *Reimer et al., 2016*) are associated with cholinergic activity in visual cortex. Overall, cholinergic activity in cortex is closely related to movement, but whether movement is the primary determinant of cholinergic activity and how movement-related activity compares to sensory driven activity is less clear.

Likely the primary functional effect of increased levels of acetylcholine in cortex is increased sensory responsiveness. Local application of acetylcholine has been shown to increase sensory responses in somatosensory cortex (*Chatfield and Dempsey, 1942*), auditory cortex (*Forster and McCarter, 1946*), as well as visual cortex (*Sato et al., 1987*) in anesthetized cats. A similar gain in visually evoked responses in rodent visual cortex can be induced by iontophoretic application of acetylcholine in cortex (*Sillito and Kemp, 1983*), by systemic acetylcholine release following basal forebrain stimulation (*Goard and Dan, 2009*; *Pafundo et al., 2016*), or by optogenetic stimulation of basal forebrain cholinergic axons in visual cortex (*Pinto et al., 2013*). Interestingly, this increase of sensory responses caused by acetylcholine is not homogeneous across cortical layers. Neurons responsive to iontophoretic application of acetylcholine are preferentially found in deep cortical layers (*Krnjevic and Phillis, 1963*). The effects of acetylcholine are primarily one of facilitation of neuronal responses in deep cortical layers and one of suppression in superficial layers (*Sillito and Kemp, 1983*; *Soma et al., 2013*). However, depending on how locally acetylcholine is applied, it can also result in selective hyperpolarization of layer 5 neurons (*Gulledge et al., 2007*). Triggering systemic increase of acetylcholine by nucleus basalis stimulation results in firing rate increases in layers 4, 5, and 6 of visual cortex, but decreased firing rates in layer 2/3 (*Goard and Dan, 2009*). While it is unclear whether the effect observed in the experiments that use systemic modulation of cholinergic activity is cortical or reflects a combination of changes in subcortical and cortical processing, a direct layer-specific effect on cortical processing would be consistent with differential expression of acetylcholine receptors across cortical layers (*Obermayer et al., 2017*). Based on these findings it is often assumed that the local release of acetylcholine in cortex differentially influences deep and superficial cortical layers.

A second functional effect of acetylcholine is a decorrelation of cortical activity. Both basal forebrain cholinergic neuron stimulation and local cholinergic axon stimulation in visual cortex desynchronizes activity in visual cortex (*Pinto et al., 2013*). This effect is thought to be mediated by the influence of acetylcholine on somatostatin-positive interneurons and their interaction with the excitatory neurons. This is based on the finding that optogenetic inhibition of somatostatin interneurons blocks acetylcholine-induced decorrelation (*Chen et al., 2015*), and that optogenetically induced acetylcholine release selectively enhances the input from excitatory neurons onto somatostatin interneurons (*Alitto and Dan, 2012*). A similar decrease in correlation has also been observed during locomotion (*Aydın et al., 2018*; *Erisken et al., 2014*), and the effect was stronger in deeper layers of cortex (*Dadarlat and Stryker, 2017*). It is possible that the decrease of correlations observed during locomotion is driven by locomotion-related increases in acetylcholine.

More generally, following the discovery of the modulation of sensory evoked responses by locomotion in visual cortex (*Niell and Stryker, 2010*), it has been speculated that acetylcholine is involved

in locomotion-related changes in neuronal activity. Based on studies in rodents, a circuit was delineated that projects from mesencephalic locomotor region to the basal forebrain, which when active, can bring about a gain of responses in visual cortex to visual stimuli (*Lee et al., 2014*). It has been argued that nicotinic activation of vasoactive intestinal peptide (VIP) expressing interneurons in visual cortex is necessary for this locomotion-induced gain of visual responses (*Fu et al., 2014*). While VIP interneurons are clearly influenced by acetylcholine (*Gasselin et al., 2021*), cholinergic receptors are expressed in all cortical cell types (*Colangelo et al., 2019*), and there are likely direct functional effects of acetylcholine on many other cell types (*Alitto and Dan, 2012*; *Chen et al., 2015*; *Gulledge and Stuart, 2005*; *Hay et al., 2016*; *Hedrick and Waters, 2015*; *Unal et al., 2012*; *Xiang et al., 1998*; *Zolles et al., 2009*). This has called into question the interpretation that acetylcholine is involved in locomotion-related changes in neuronal activity primarily through nicotinic activation of VIP neurons - it is more likely that most cortical cell types are directly influenced (*Pakan et al., 2016*). The specific subtypes of muscarinic and nicotinic receptors expressed across these different cell types further complicate the causal attribution of effects. Nevertheless, all these lines of evidence come together to suggest that basal forebrain cholinergic axons in visual cortex increase their activity on locomotion and release acetylcholine to bring about a gain of visually evoked responses. Throughout the manuscript, we will refer to basal forebrain cholinergic axons simply as cholinergic axons. This distinction notably excludes potential axons that arise from local VIP-ChAT$^+$ neurons (*Granger et al., 2020*), that are labeled when using a cross of ChAT-Cre mice with a reporter line. Prior measurements of the activity of cholinergic axons in visual cortex all either have relied on data from a cross of ChAT-Cre mice with a reporter line (*Larsen et al., 2018*; *Reimer et al., 2016*), which resulted in co-labeling of local VIP-ChAT$^+$ neurons, or have pooled imaging data across cortical regions (*Collins et al., 2023*), which obscures the projection-specificity of basal forebrain cholinergic axons to cortex (*Kim et al., 2016*). Thus, it remains unclear what the response profile of cholinergic axons in mouse visual cortex is and what the functional role of this input may be.

In this work, we sought to understand the signals conveyed by the basal forebrain cholinergic system to visual cortex, the consequent changes to the activity of layer 2/3 and layer 5 neurons in visual cortex upon release of acetylcholine, and the potential implications this could have on the computations performed by visual cortex. We used two-photon calcium imaging of cholinergic axons in visual cortex in behaving head-fixed mice and found that cholinergic activity in visual cortex was best approximated as a locomotion state signal. The effects of optogenetic activation of cholinergic axons locally in visual cortex were layer-specific: the responsiveness was increased in neurons in layer 5 but not in layer 2/3. Furthermore, we provide evidence that acetylcholine release underlies the locomotion-related decorrelation of activity in visual cortex.

## Results

To investigate the function of cholinergic input to visual cortex, we first characterized the calcium activity in these axons. We used an AAV vector to express an axon-targeted GCaMP6s (*Broussard et al., 2018*) in basal forebrain cholinergic neurons using ChAT-IRES-Cre mice (*Rossi et al., 2011*) and recorded calcium activity of their axons in visual cortex using two-photon microscopy (*Figure 1A and B*). For imaging experiments, mice were head-fixed on a spherical treadmill surrounded by a toroidal screen (*Figure 1C*). We used a set of different visuomotor conditions known to activate neurons in visual cortex to probe for activation of cholinergic axons. First, mice were exposed to a closed loop condition during which locomotion velocity was coupled to movement in a virtual tunnel. We then measured activity in an open loop condition during which locomotion and movement in the virtual tunnel were uncoupled, in darkness, and during the presentation of full field drifting gratings. Throughout all experimental conditions mice were free to locomote on the spherical treadmill and did so over a range of velocities (*Video 1*).

We found that the activity in many cholinergic axons strongly increased during locomotion (*Figure 1D and E*). Activity increased with locomotion onset (*Figure 1F*) with an average lag across all responsive axons of 477 ms relative to locomotion onset (*Figure 1—figure supplement 1A*). Thus, consistent with results reported for auditory cortex (*Nelson and Mooney, 2016*), we found that locomotion preceded the increase in cholinergic activity in visual cortex. Locomotion onset also results in an increase in activity of neurons in primary visual cortex (*Ayaz et al., 2013*; *Keller et al., 2012*; *Saleem et al., 2013*). It has been speculated that cholinergic input is a driver of locomotion-related

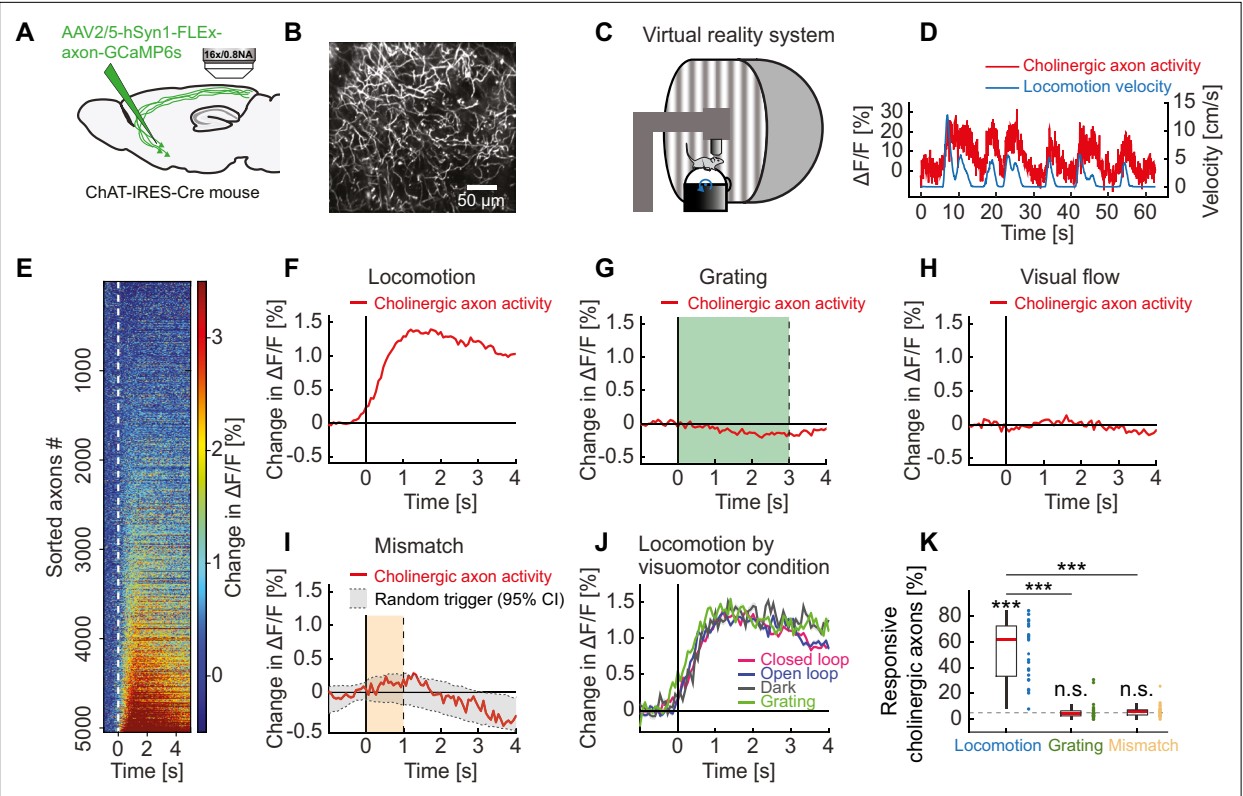

**Figure 1.** Cholinergic axons in visual cortex are activated during locomotion independent of visual stimuli. (**A**) GCaMP6s was expressed by injecting an AAV2/5-hSyn1-FLEx-axon-GCaMP6s vector in basal forebrain of ChAT-IRES-Cre mice. We then imaged calcium activity of cholinergic axons projecting to visual cortex. (**B**) Example two-photon image of cholinergic axons in visual cortex. (**C**) Schematic of the virtual reality system. Mice were head-fixed on a spherical treadmill and surrounded by a toroidal screen on which we presented visual stimuli in different visuomotor conditions. (**D**) Example trace of the calcium activity of one cholinergic axon in visual cortex (red) and the corresponding locomotion velocity of the mouse (blue). (**E**) Average locomotion onset activity of all cholinergic axons (5048 axons in 14 mice), sorted by their average response during locomotion onset. (**F**) Average response of cholinergic axons to locomotion onset over all the cholinergic axons, of the data shown in **E**. Shading indicates SEM over axons. Due to SEM being very small, the shading is fully obscured by the red line. (**G**) As in **F**, but for grating onsets. Green shading marks the duration of grating presentation. (**H**) As in **F**, but for visual flow onset during the open loop condition. (**I**) As in **F**, but for visuomotor mismatch onset. Orange shading marks the duration of visuomotor mismatch. As mismatch events occur only during times of locomotion, and locomotion itself is a strong driver of cholinergic activity, we would expect to find an increase in cholinergic activity by chance at mismatch. To correct for this, we quantified the distribution of cholinergic activity on random triggers during locomotion (95% confidence interval (CI), gray shading). (**J**) Locomotion onset activity in different visuomotor conditions. (**K**) The fraction of cholinergic axons responsive to locomotion, grating, and visuomotor mismatch onset, quantified for each imaging site. Each datapoint is one imaging site. Boxes show 25th and 75th percentile, central mark is the median, and the whiskers extend to the most extreme data points not considered outliers. Dashed line marks chance level. n.s.: not significant; *p<0.05; **p<0.01; ***p<0.001; see *Supplementary file 1* for all statistical information.

The online version of this article includes the following figure supplement(s) for figure 1:

**Figure supplement 1.** Cholinergic axons in visual cortex were activated after layer 2/3 but before layer 5 visual cortex neurons.

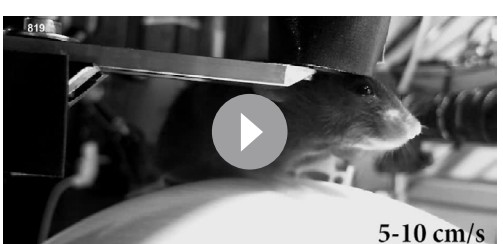

**Video 1.** Mouse running speeds. Mouse running on a spherical treadmill at different speeds to illustrate the range of running speeds covered by our data.
https://elifesciences.org/articles/89986/figures#video1

activity in visual cortex (*Lee et al., 2014*). We thus compared the observed response latency between cholinergic input and activity of layer 2/3 and layer 5 neurons. The response latency of cholinergic axons was longer than the latency we observed in responses of layer 2/3 neurons (*Figure 1—figure supplement 1B*), but shorter than that of layer 5 neurons (*Figure 1—figure supplement 1C, D*). This would suggest that the initial increase in neuronal activity in layer 2/3 of the visual cortex was not driven by cholinergic input from basal forebrain.

Interestingly, we found no evidence of a response of cholinergic axons to other stimuli that drive responses in visual cortex such as full field drifting gratings (*Figure 1G*), visual flow onsets in open loop (*Figure 1H*), or visuomotor mismatches (*Figure 1I*; *Attinger et al., 2017*; *Jordan and Keller, 2020*; *Keller et al., 2012*; *Niell and Stryker, 2008*; *Vasilevskaya et al., 2023*; *Widmer et al., 2022*). Visuomotor mismatches are brief visual flow halts that break the coupling between locomotion and visual flow feedback in a closed loop condition. Consistent with the absence of response to visual stimuli – full-field drifting gratings or visual flow in open loop, we found that the response to locomotion onset was independent of whether the locomotion occurred in darkness, in the presence of visual stimuli, or with visual flow feedback coupled to locomotion (*Figure 1J*). For each imaging site, we then quantified the percentage of cholinergic axons that exhibited significant responses to locomotion onset, full field drifting gratings, or visuomotor mismatch, and found that 52.3% ± 4.4% (mean ± SEM) of the axons responded significantly to locomotion onset, while the fraction of responsive axons to gratings and mismatch was not different from chance (*Figure 1K*).

To confirm that an increase in calcium activity in cholinergic axons corresponds to an increase in extracellular acetylcholine, we measured acetylcholine levels using the genetically-encoded

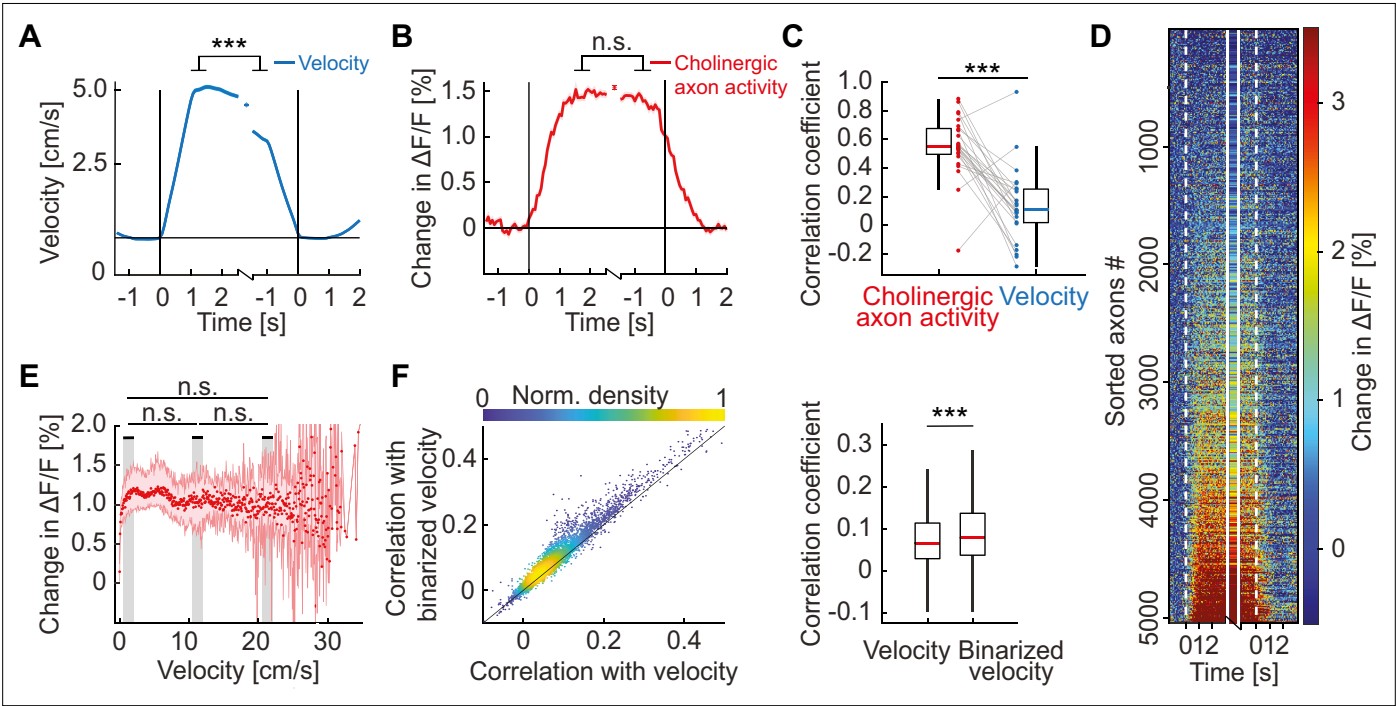

**Figure 2.** Activity in cholinergic axons was better explained by a binary locomotion state signal than by locomotion velocity. (**A**) Average locomotion velocity profiles aligned to locomotion onset and offset. The isolated datapoint between the two traces is the average locomotion velocity over the locomotion bout. Shading marks SEM. Here and elsewhere, n.s.: not significant; *p<0.05; **p<0.01; ***p<0.001; see *Supplementary file 1* for all statistical information. (**B**) As in **A**, but for activity of all cholinergic axons in visual cortex, for the data shown in **D**. Shading marks SEM over axons. (**C**) Bout-by-bout correlation coefficient between the locomotion onset and offset changes for average cholinergic activity and locomotion velocity. Each datapoint is one imaging site. Boxes show 25th and 75th percentile, central mark is the median, and the whiskers extend to the most extreme data points not considered outliers. (**D**) Average onset and offset responses for all cholinergic axons sorted by strength of locomotion onset response. (**E**) Activity of cholinergic axons as a function of locomotion velocity. Shown are the hierarchical bootstrap estimates of the median (red dots) and 95% confidence intervals (red shading) for each velocity bin. We found no evidence of a difference in activation between low, intermediate, and high locomotion velocities (indicated by gray shading). (**F**) Left: For all axons, the correlation coefficient of their calcium activity with locomotion velocity plotted against the correlation coefficient with a binarized version of the locomotion velocity. Right: Distributions of correlation coefficients between calcium activity and locomotion velocity and binarized velocity. Same data as shown on the left. Boxes show 25th and 75th percentile, central mark is the median, and the whiskers extend to the most extreme data points not considered outliers.

The online version of this article includes the following figure supplement(s) for figure 2:

**Figure supplement 1.** Responses of GRAB-ACh3.0 in visual cortex are similar to the calcium responses in cholinergic axons.

**Figure supplement 2.** Cholinergic axon activity correlated better with a binarized locomotion vector than with the unfiltered locomotion vector over a large range of binarization thresholds.

acetylcholine sensor GRAB-Ach3.0 (*Jing et al., 2020*). The sensor was expressed in visual cortex, and we performed two-photon imaging in layer 2/3 during the same visuomotor conditions as described above. We found that consistent with the calcium activity of cholinergic axons, acetylcholine levels in visual cortex increased on locomotion onsets (*Figure 2—figure supplement 1A*), while grating onsets (*Figure 2—figure supplement 1B*) and visuomotor mismatches (*Figure 2—figure supplement 1C*) resulted in no measurable increase when corrected for hemodynamic artifacts (*Figure 2—figure supplement 1A–F*). Thus, consistent with previous reports (*Larsen et al., 2018*), we find that locomotion is associated with a basal forebrain-driven increase in acetylcholine levels in visual cortex.

Further characterizing the relationship between locomotion velocity and cholinergic activity, we found that cholinergic axon activity was not linearly related to locomotion velocity, but instead exhibited a locomotion state dependence. Over the course of a locomotion bout, we found that velocity decreased systematically, such that towards the end of a locomotion bout, velocity was lower than at the beginning (*Figure 2A*). For cholinergic activity, this decrease was absent. We found no evidence of a difference in the average calcium activity of cholinergic axons after locomotion onset to that before locomotion offset (*Figure 2B*). Indeed, the average cholinergic activity after locomotion onset and before locomotion offset was highly correlated across locomotion bouts, while a similar analysis on locomotion velocity revealed that the time-course of velocity is more variable and less correlated bout-by-bout (*Figure 2C*). The similarity of locomotion onset and offset responses was not only true for the population average activity, but the activity after locomotion onset, before locomotion offset, and the average activity during the locomotion bout were all well correlated on an axon-by-axon level (*Figure 2D* and *Figure 2—figure supplement 1G*). To quantify the relationship between locomotion velocity and calcium activity in cholinergic axons more generally, we computed the average calcium activity as a function of locomotion velocity (see Methods). This relationship was more step-like than linear, and we found no evidence of an increase in calcium activity between low and high locomotion velocities (*Figure 2E*). The same held true for extracellular acetylcholine levels as measured by GRAB-ACh3.0 (*Figure 2—figure supplement 1H*). If calcium activity in cholinergic axons reflected a step-like change during locomotion, we should find that calcium activity correlates better with a binarized version of locomotion velocity than with locomotion velocity itself. This is indeed what we found (*Figure 2F*), independent of the value of the threshold used for binarization (*Figure 2—figure supplement 2*). Thus, calcium activity in cholinergic axons in visual cortex is better described as a binary locomotion state signal, than a linear locomotion velocity signal.

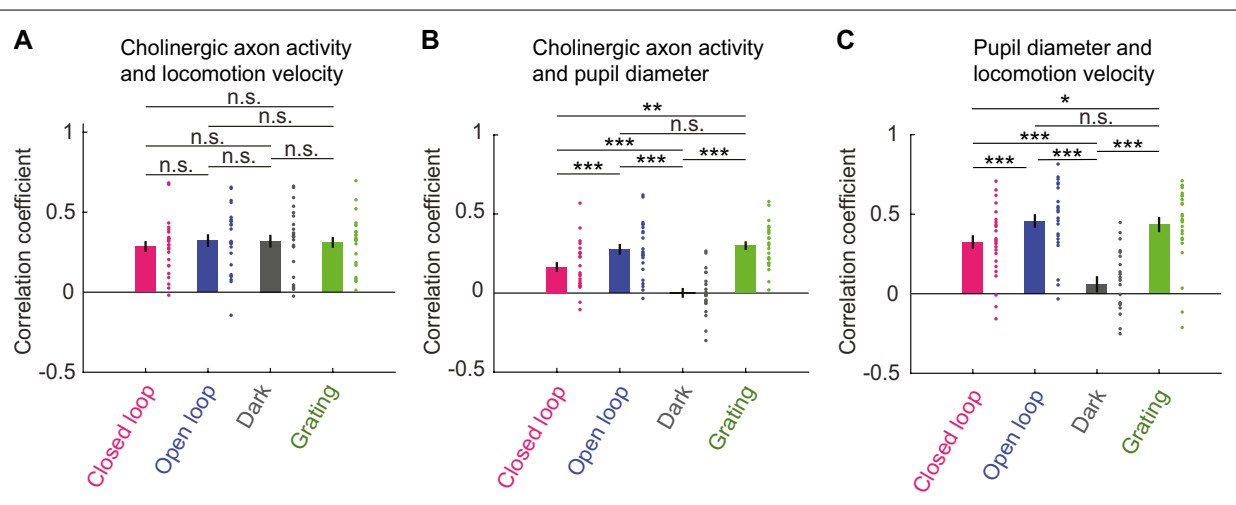

**Figure 3.** Average activity of cholinergic axons was more strongly correlated with locomotion velocity than with pupil diameter. (**A**) Correlation of average activity of cholinergic axons activity with locomotion velocity in closed loop, open loop, dark, and grating conditions. Each point represents data from one imaging site, error bars indicate SEM. Here and elsewhere, n.s.: not significant; *p<0.05; **p<0.01; ***p<0.001; see *Supplementary file 1* for all statistical information. (**B**) As in **A**, but for correlation between average activity of cholinergic axons and pupil diameter. (**C**) As in **A**, but for correlation between locomotion velocity and pupil diameter.

The online version of this article includes the following figure supplement(s) for figure 3:

**Figure supplement 1.** Activity of cholinergic axons correlated better with locomotion velocity than with facial movement.

A number of behavioral variables, like pupil dilation, facial movements, as well as the overall level of arousal are all correlated with locomotion (*Lohani et al., 2022*; *Reimer et al., 2014*; *Vinck et al., 2015*). It has been shown that cholinergic activity in visual cortex is also well correlated with pupil dilation and facial movements (*Larsen et al., 2018*; *Lohani et al., 2022*; *Reimer et al., 2014*). Disambiguating the effects of locomotion, pupil dilation, and intensity of facial movements on cholinergic activity is complicated by the fact that these three variables are strongly correlated during behavior. It has been shown that under certain conditions, facial movements are a better predictor of cholinergic activity than locomotion across large parts of dorsal cortex (*Lohani et al., 2022*). To quantify this relationship in visual cortex, we compared the correlation between cholinergic activity, locomotion velocity, pupil diameter, and intensity of facial movements in different visuomotor conditions. This analysis was motivated by our observation that average light levels have a stronger influence on pupil size than locomotion state. We found that the correlation of locomotion velocity and cholinergic activity was independent of visuomotor condition (*Figure 3A*). This was not the case for the correlation between pupil diameter and cholinergic activity, which was systematically lower in darkness than in conditions with visual stimuli, and lower during closed loop visuomotor coupling than during open loop or grating conditions (*Figure 3B*). This pattern was also reflected in the correlation between locomotion and pupil diameter (*Figure 3C*). The reduced correlation between locomotion velocity and pupil diameter in darkness was not a consequence of a decrease in variability due to ceiling-effects in pupil diameter in darkness (*Figure 3—figure supplement 1A*). Finally, we also computed the correlation between cholinergic activity and facial movements. The correlation of cholinergic activity with locomotion was higher than that with facial movements under all conditions that we tested (*Figure 3—figure supplement 1B*). Thus, of these three behavioral variables, locomotion, pupil diameter, and facial movements, locomotion exhibited the most robust correlation with cholinergic activity in visual cortex (*Figure 3—figure supplement 1C*).

Next, we turned to the question of what the influence of increased cholinergic activity in visual cortex is on the activity of neurons in layer 2/3 and layer 5. We know that locomotion influences neuronal activity in visual cortex in a variety of ways. One of these effects is that during locomotion, visual responses in visual cortex are increased (*Niell and Stryker, 2010*) through a combination of both additive and multiplicative effects (*Dadarlat and Stryker, 2017*). Consistent with this we found locomotion driven increases in visual responses were apparent in both layer 2/3 (*Figure 4A*) and in layer 5 neurons (*Figure 4B*). To test whether this increase in visual responsiveness during locomotion was driven by acetylcholine, we expressed the excitatory opsin ChrimsonR *Klapoetke et al., 2014* in basal forebrain cholinergic neurons using an AAV2/1-hSyn-DIO-ChrimsonR-tdTomato vector in ChAT-IRES-Cre mice. We injected an AAV2/1-Ef1α-GCaMP6f in visual cortex to express GCaMP6f. In cortex, the Ef1α promoter drives highly specific expression in neurons (99.7% of labeled cells in layer 2/3 and 100% of labeled cells in layer 5 are NeuN-positive *Yaguchi et al., 2013*), and biases expression to excitatory neurons, such that approximately 95% of labeled neurons are excitatory in both layer 2/3 (*Attinger et al., 2017*; *Yaguchi et al., 2013*) and in layer 5 (*Yaguchi et al., 2013*). We then performed two-photon calcium imaging in visual cortex to measure the responses of layer 2/3 or layer 5 neurons to the optogenetic activation of local cholinergic axons paired to the presentation of full field drifting grating stimuli while the mouse was stationary (see Methods). We found that in layer 2/3, optogenetic activation of cholinergic axons did not result in a detectable increase in grating onset responses (*Figure 4C* and *Figure 4—figure supplement 2A*), while the responses of layer 5 neurons to the same stimulus increased with concurrent optogenetic activation of cholinergic axons (*Figure 4D*). The effects of optogenetic activation of cholinergic axons on layer 5 neurons could not be explained by a stimulation-triggered change in locomotion velocity (*Figure 4—figure supplement 1*) or stimulation artifacts (*Figure 4—figure supplement 3A–C*). In mice that did not express ChrimsonR in cholinergic axons (no ChrimsonR controls), we found no evidence of an optogenetic light stimulation effect. Interestingly, this increase in grating onset responses in layer 5 neurons driven by direct cholinergic activation was smaller than that observed during locomotion (*Figure 4B and D*). This could be either because the acetylcholine release triggered by the optogenetic stimulation of cholinergic axons was weaker than that occurring during locomotion, or because the increase in responses during locomotion is only partially explained by the co-release of acetylcholine. If the former were the case, we would expect that responses occurring during locomotion are less influenced by optogenetic activation of cholinergic axons. This was indeed the case for grating onset responses during locomotion which

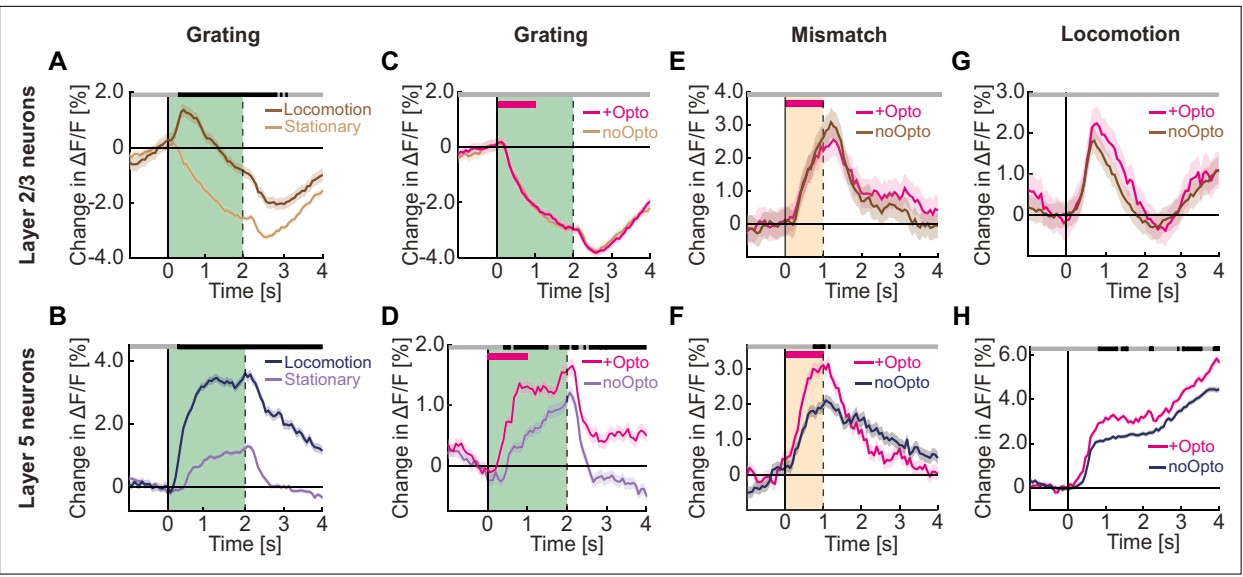

**Figure 4.** Optogenetic activation of cholinergic axons in visual cortex primarily enhances responses of layer 5, but not layer 2/3 neurons. (**A**) Average calcium response of layer 2/3 neurons in visual cortex to full field drifting gratings in the absence (light brown) or presence of locomotion (brown). Green shading indicates duration of grating stimulus. The responses are compared bin-by-bin using a nested hierarchical bootstrap test. Here and in subsequent panels, bins with a significant difference (p<0.05) are indicated by a black line above the plot; those with p>0.05 are marked gray. Shading marks SEM. (**B**) As in **A**, but for layer 5 neurons in visual cortex in the absence (light blue) or presence (dark blue) of locomotion. (**C**) Average calcium response of layer 2/3 neurons in visual cortex to full field drifting gratings while the mice were stationary, without (light brown) or with (pink) optogenetic activation of cholinergic axons in visual cortex. Duration of optogenetic stimulation is marked by a pink bar. Green shading indicates duration of the grating stimulus. (**D**) As in **C**, but for layer 5 neurons in visual cortex. (**E**) Average calcium response to visuomotor mismatch in layer 2/3 neurons in visual cortex, without (brown) and with (pink) optogenetic stimulation of cholinergic axons in visual cortex. Orange shading indicates mismatch duration, and pink bar indicates optogenetic stimulation. (**F**) As in **E**, but for layer 5 neurons in visual cortex. (**G**) Average calcium response of layer 2/3 neurons in visual cortex to locomotion onset in closed loop, without (brown) or with (pink) optogenetic stimulation of cholinergic axons in visual cortex. (**H**) As in **G**, but for layer 5 neurons in visual cortex.

The online version of this article includes the following figure supplement(s) for figure 4:

**Figure supplement 1.** Optogenetic stimulation of basal forebrain cholinergic axons in visual cortex did not change the locomotion velocity of the mice.

**Figure supplement 2.** Optogenetic activation of cholinergic axons in visual cortex does not increase the responsiveness of the layer 2/3 neurons most responsive to gratings, visuomotor mismatch, or locomotion onset.

**Figure supplement 3.** Optogenetic activation did not increase grating responses during locomotion and had no effect in opsin-negative control mice.

**Figure supplement 4.** Cholinergic stimulation primarily has a multiplicative influence on the orientation tuning curve of layer 5 neurons.

**Figure supplement 5.** Locomotion onset responses are suppressed by closed loop visual feedback in layer 2/3, but not layer 5.

were not modulated by optogenetic activation of cholinergic axons (*Figure 4—figure supplement 3D, E*). Interestingly, visuomotor mismatch responses, which also occur only during locomotion, were significantly increased in layer 5 neurons (*Figure 4F*), but not in layer 2/3 (*Figure 4E* and *Figure 4—figure supplement 2B*). Similarly, we found that optogenetic activation of cholinergic axons increased locomotion onset responses in layer 5 neurons (*Figure 4H*), while layer 2/3 neurons exhibited no significant difference (*Figure 4G* and *Figure 4—figure supplement 2C*). Thus, while in layer 5 neurons the effect of optogenetic activation appears to saturate for bottom-up grating responses during locomotion, this was not the case for responses that rely on top-down input. However, it is also likely that the increase in response during locomotion is only partially explained by acetylcholine. Locomotion has been shown to result in both a multiplicative and an additive change in grating responses (*Dadarlat and Stryker, 2017*). This was also the case in our data for locomotion-related increase in grating responses (*Figure 4—figure supplement 4*). However, we found that the optogenetic stimulation of cholinergic axons resulted primarily in a multiplicative gain of visual responses (*Figure 4—figure supplement 4*). Overall, consistent with previous findings (*Goard and Dan, 2009*), we find that acetylcholine primarily increases both bottom-up and top-down driven responses in layer 5 but not layer 2/3 neurons, but only partially explains the locomotion-driven increases in visual responses.

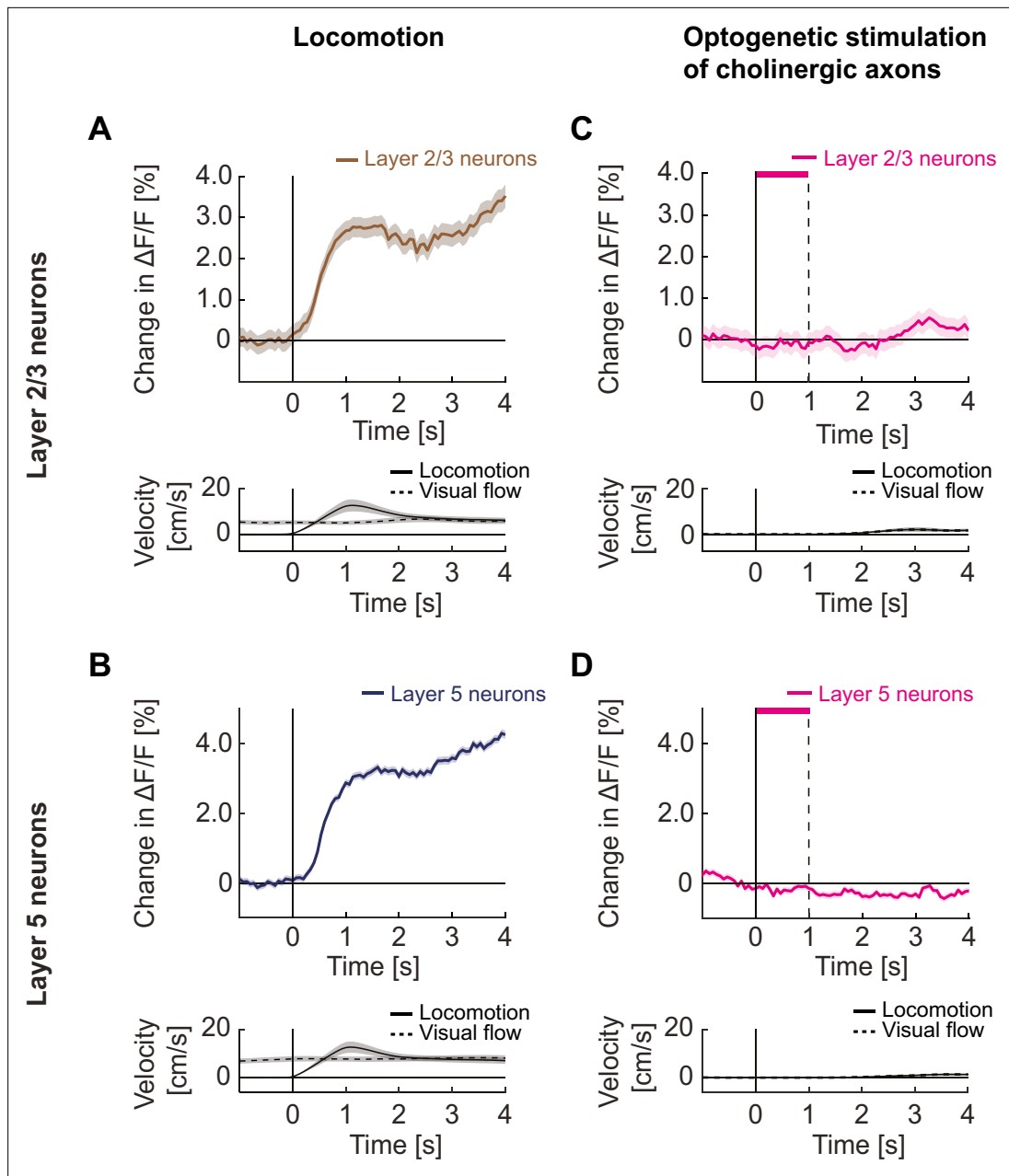

**Figure 5.** Optogenetic activation of cholinergic axons in visual cortex did not drive responses in either layer 2/3 or layer 5 neurons. (**A**) Upper panel: Average calcium response of layer 2/3 neurons in visual cortex to locomotion onset in open loop. Lower panel: Average locomotion velocity and visual flow speed for the corresponding onsets. Shading indicates SEM. (**B**) As in **A**, but for layer 5 neurons in visual cortex. (**C**) Upper panel: Average calcium response of layer 2/3 neurons in visual cortex during local optogenetic stimulation of cholinergic axons while the mice were stationary. Pink bar marks duration of the optogenetic stimulus. Lower panel: Average locomotion velocity and visual flow speed for the corresponding onsets. Note, the two overlap by experimental design. (**D**) As in **C**, but for layer 5 neurons in visual cortex.

A second effect of locomotion on neuronal activity in visual cortex is that locomotion itself drives neuronal responses (*Keller et al., 2012*; *Saleem et al., 2013*). This effect is apparent in both layer 2/3 (*Figure 5A*) and layer 5 (*Figure 5B*) neurons. In layer 2/3, locomotion-related activity is suppressed by closed loop visual feedback (*Widmer et al., 2022*), as a function of the strength of visuomotor mismatch responses (*Figure 4—figure supplement 5A, B*). In layer 5, closed loop visual feedback did not suppress running-related activity on average in either visuomotor mismatch or grating responsive neurons (*Figure 4—figure supplement 5C, D*). Thus, locomotion related activity likely has functionally distinct roles in layer 2/3 and layer 5. To test whether artificial activation of cholinergic axons

would result in increased neuronal activity, we again performed two-photon calcium imaging in visual cortex neurons to measure responses of layer 2/3 or layer 5 neurons upon optogenetic activation of local cholinergic axons. We found no evidence of any increase in calcium activity in either layer 2/3 (*Figure 5C*) or layer 5 neurons (*Figure 5D*). This absence of a stimulation response is consistent with previous reports that found that cholinergic axon stimulation in visual cortex does not result in a response of layer 2/3 neurons (*Chen et al., 2015*) and that electrical stimulation of the basal forebrain primarily activates inhibitory neurons in the superficial layers of visual cortex (*Alitto and Dan, 2012*). This, combined with our finding that on average the layer 2/3 locomotion onset-related activity preceded the activity increase in cholinergic axons (*Figure 1—figure supplement 1*), would indicate that the locomotion-related increase in layer 2/3 and layer 5 activity observed in visual cortex is not driven by acetylcholine.

A third effect locomotion has on neuronal activity in visual cortex is to decorrelate the activity of nearby neurons (*Dadarlat and Stryker, 2017; Eriksen et al., 2014*). Consistent with these studies, we

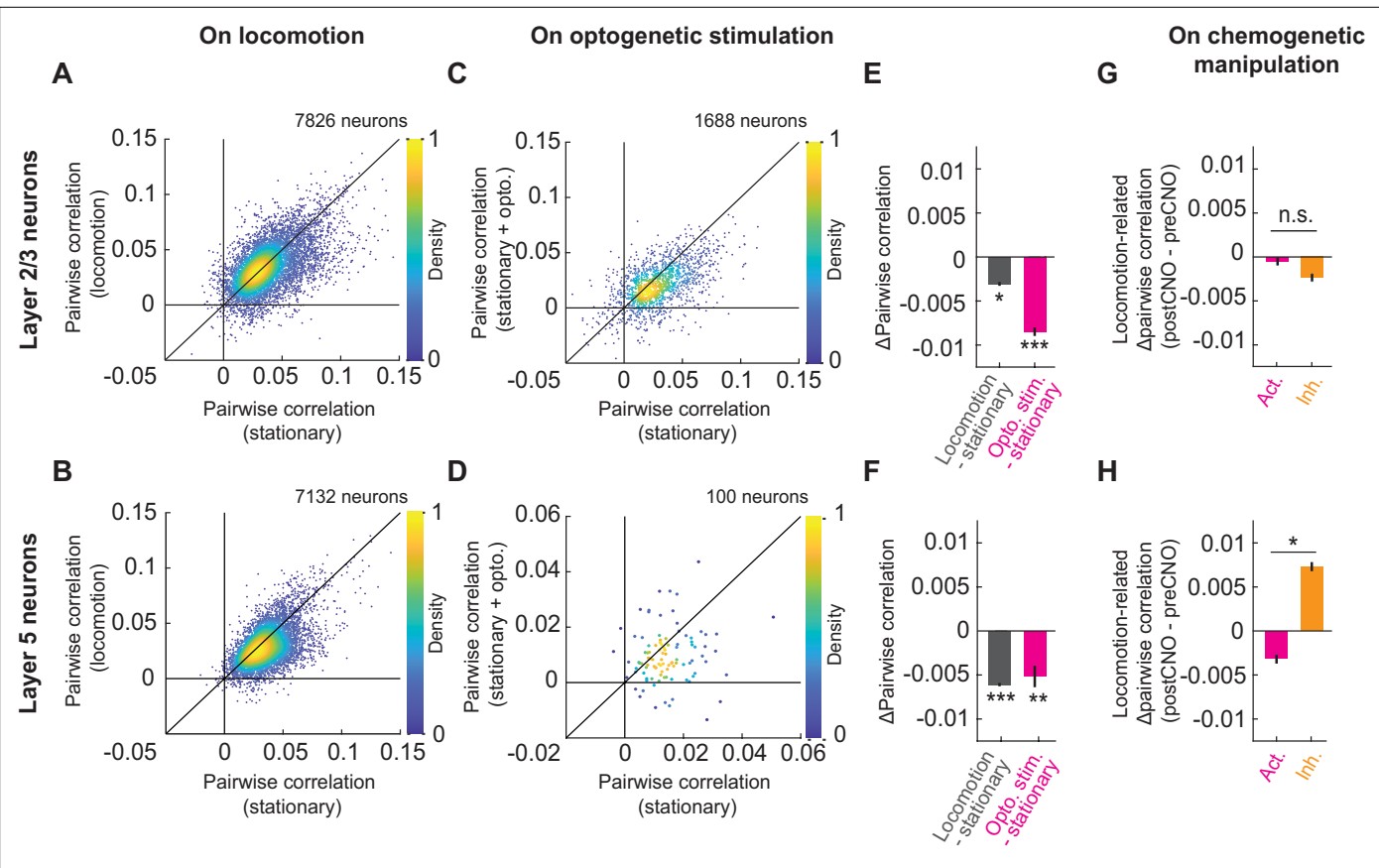

**Figure 6.** Cholinergic axon activation reduced pairwise correlations between neurons in visual cortex. (**A**) Average pairwise correlation in calcium activity of layer 2/3 neurons while the mouse was stationary and while the mouse was locomoting. Each dot is the mean of the pairwise correlations for one neuron to all others in the same field of view. (**B**) As in **A**, but for layer 5 neurons. (**C**) As in **A**, but comparing the correlation while the mice were stationary, to that while cholinergic axons were stimulated optogenetically. (**D**) As in **C**, but for layer 5 neurons. Note, for this experiment the imaging plane was in L2/3 to simultaneously image layer 2/3 neurons and apical dendrites of layer 5. The latter were traced to their soma in layer 5 offline and were used to compute layer 5 activity correlations. (**E**) Average change in pairwise correlations of layer 2/3 neurons on locomotion (gray) or on optogenetic stimulation of cholinergic axons during stationary periods (pink), compared to the correlations during stationary (stat.) periods at baseline. Here and elsewhere, n.s.: not significant; *p<0.05; **p<0.01; ***p<0.001; see *Supplementary file 1* for all statistical information. (**F**) As in **E**, but for layer 5 neurons of visual cortex. (**G**) Change in the average pairwise correlations of layer 2/3 neurons during locomotion, upon DREADD activation (pink) or inhibition (orange) of basal forebrain cholinergic neurons. (**H**) As in G, but for layer 5 neurons.

The online version of this article includes the following figure supplement(s) for figure 6:

**Figure supplement 1.** Locomotion increases the correlation between those layer 5 neurons in visual cortex that on average have a higher correlation.

**Figure supplement 2.** Chemogenetic manipulation of basal forebrain cholinergic neurons affects locomotion behavior of mice.

**Figure supplement 3.** Change in locomotion-related decorrelation upon chemogenetic manipulation of basal forebrain cholinergic neurons.

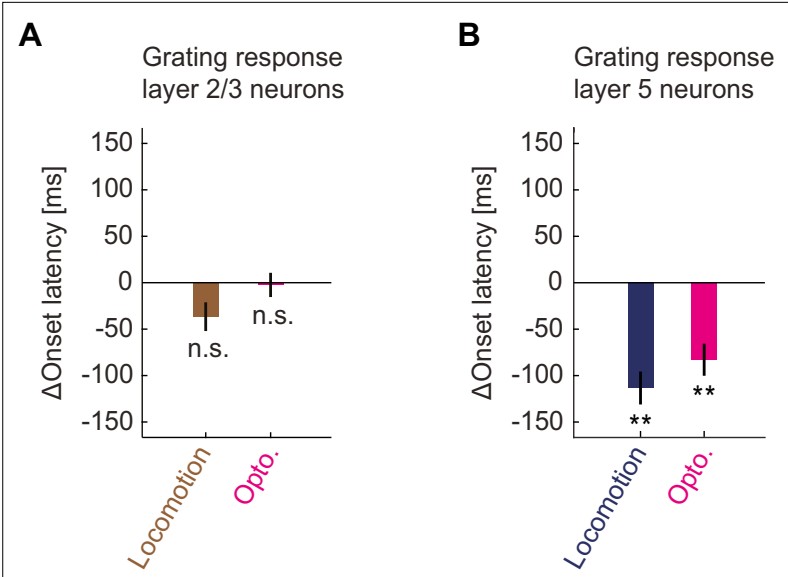

**Figure 7.** Cholinergic input reduced response latency of layer 5 neurons in visual cortex to grating stimuli. (**A**) Locomotion (brown) or optogenetic stimulation of cholinergic axons (pink) induced change in the response latency of layer 2/3 neurons in visual cortex to full field drifting grating onset. Here and elsewhere, n.s.: not significant; *p<0.05; **p<0.01; ***p<0.001; see *Supplementary file 1* for all statistical information. (**B**) As in **A**, but for layer 5 neurons.

The online version of this article includes the following figure supplement(s) for figure 7:

**Figure supplement 1.** Locomotion and optogenetic stimulation of basal forebrain cholinergic axons decreased the latency of response to a visual stimulus in layer 5 neurons but not in layer 2/3 neurons.

found that locomotion results in a reduction of pairwise correlations between the activity of layer 2/3 neurons (*Figure 6A and E*) as well as between layer 5 neurons (*Figure 6B and F*). We could induce this decorrelation by optogenetically activating the cholinergic axons while mice were stationary. This optogenetically induced decorrelation in layer 2/3 (*Figure 6C and E*) was stronger than that driven by locomotion, while the two were similar in layer 5 (*Figure 6D and F*). Note, we used calcium activity of apical dendrites of L5 neurons in L2/3 as a proxy for somatic L5 activity in these experiments (see Methods). This decorrelation was not due to a visual response to the stimulation laser (*Figure 4— figure supplement 3F*). Interestingly, we found that in both layer 2/3 and layer 5, the locomotion-induced decorrelation was observed primarily for neurons with low pairwise correlations. Pairs of neurons with high correlation of activity tended to further increase their correlation during locomotion (*Figure 6—figure supplement 1*). Finally, we tested whether the correlation in visual cortex was similarly influenced by chemogenetic manipulation of basal forebrain cholinergic neurons. We expressed either a DREADD activator or a DREADD inhibitor in ChAT-Cre mice to systemically increase or decrease cholinergic activity. Interestingly, we noticed that the systemic increase or decrease of cholinergic release resulted in behavioral changes in the mice. Upon chemogenetic activation mice increased their locomotion velocity, and upon chemogenetic inhibition, mice decreased their time spent locomoting, resulting in a bidirectional modulation of the total distance traveled by the two manipulations (*Figure 6—figure supplement 2*). Importantly, chemogenetic manipulations bidirectionally modulated the decorrelation effect on locomotion in layer 5 neurons in visual cortex. DREADD activation resulted in an increased strength of the locomotion-driven decorrelation, while DREADD inhibition resulted in a reduced strength of locomotion-driven decorrelation in layer 5 (*Figure 6H* and *Figure 6—figure supplement 3B*). This effect was absent in layer 2/3 where neither manipulation influenced the locomotion driven decorrelation (*Figure 6G* and *Figure 6—figure supplement 3A*). Thus, cholinergic activation is likely sufficient to explain the locomotion-induced decorrelation of activity in visual cortex.

Finally, an effect of locomotion on the responses of layer 5 neurons, which to the best of our knowledge is unreported, is a decrease in latency to response. In layer 5 neurons, responses to grating stimuli

appeared earlier during locomotion than they did while the mouse was stationary (*Figures 4B and 7B*). This decrease in response latency during locomotion was absent in layer 2/3 neurons (*Figures 4A and 7A*). Optogenetic activation of cholinergic axons was able to recapitulate the decrease in response latency in layer 5 neurons (*Figures 4D and 7B* and *Figure 7—figure supplement 1B*), while leaving layer 2/3 neuron response latency unaffected (*Figures 4C and 7A* and *Figure 7—figure supplement 1A*). Thus, cholinergic input decreases the response latency of layer 5 neurons.

## Discussion

We set out to address the question of why acetylcholine is released in visual cortex by answering two simpler, intermediate questions: what drives acetylcholine release, and what immediate effects does the released acetylcholine have on the cortical circuits. We found that locomotion was the main driver of ChAT axons activity (*Figure 1*) and acetylcholine release (*Figure 2—figure supplement 1*) in visual cortex, and cholinergic activity depended on locomotion velocity in a non-linear, step-like manner (*Figure 2*). The acute effects of this increase in acetylcholine on cortical responses were layer-specific: acetylcholine release enhanced responses specifically in layer 5 neurons, but not in layer 2/3 (*Figure 4*). Using chemogenetic manipulations of basal forebrain cholinergic neurons, we found that we could bidirectionally influence the activity correlations of layer 5 neurons. Cholinergic activation resulted in a decrease of pairwise correlations while inhibition resulted in an increase (*Figure 6H*). We observed a similar decrease in pairwise correlations during optogenetic stimulation of cholinergic axons in the visual cortex, arguing that this effect is, at least in part, mediated locally (*Figure 6F*). Similarly, optogenetic stimulation of cholinergic axons in visual cortex drove a decrease in pairwise correlation of layer 2/3 (*Figure 6E*), while global manipulation of the basal forebrain cholinergic system using chemogenetic tools did not significantly change correlations on activation or inhibition (*Figure 6G*). Our results would be consistent with the interpretation that the acute effects of acetylcholine are stronger in layer 5 where they act to increase responsiveness to inputs from outside of the local layer 5 network.

While the optogenetic activation of cholinergic axons in visual cortex likely recapitulates the local effects of acetylcholine more directly than those observed with basal forebrain stimulation, the method comes with the following caveats. First, we don't know how the pattern of acetylcholine release induced by optogenetic activation of cholinergic axons compares to that observed endogenously during locomotion. The pattern of cholinergic axon activation was not uniform across axons (*Figures 1E and 2D*), consistent with different modes of activity in cholinergic subpopulations observed in basal forebrain (*Laszlovszky et al., 2020*). It is unclear whether the exact pattern of cholinergic axon activation substantially influences the observed effects in cortex. However, given that the effects on changes in the correlation of activity induced by optogenetic activation of cholinergic axons was at least as strong as that observed during locomotion, we estimate that optogenetic activation of cholinergic axons can induce levels and patterns of acetylcholine release similar to those observed endogenously. Second, the activation of cholinergic axons likely antidromically activates cholinergic neuron soma in basal forebrain. Thus, we cannot exclude the possibility of a systemic contribution to the effects we observe through shared projections between different cortical and subcortical targets. However, the projection specificity of basal forebrain cholinergic neurons (*Kim et al., 2016*; *Pinto et al., 2013*) would argue against this. Finally, it is important to note that here we labeled basal forebrain cholinergic neurons with targeted viral injection. Thus, we cannot exclude the possibility that our labeling of cholinergic axons was incomplete. The alternative approach to label these neurons by crossing ChAT-Cre mice to a reporter line expressing a calcium indicator comes with the caveat of the presence of a substantial fraction of VIP interneurons in visual cortex that also are ChAT positive (*Granger et al., 2020*). Being unable to separate basal forebrain axons from those of local VIP interneurons complicates the interpretation of results.

Another caveat of our conclusions is the fact that our neuronal populations in both layer 2/3 and layer 5 are not random subsets. While the Ef1α promoter is specific to neurons (*Tsuchiya et al., 2002*; *Yaguchi et al., 2013*) and predominantly labels excitatory neurons (*Attinger et al., 2017*; *Yaguchi et al., 2013*), the expression levels are likely not equal across the different excitatory neuron types. In layer 5, for example, we are likely undersampling Tlx3-positive layer 5 neurons. Tlx3-positive layer 5 neurons exhibit a visually driven reduction in activity and consequently have a systematic difference between open and closed loop locomotion onsets (*Heindorf and Keller, 2022*). We do not see either

a visually driven inhibition on average (*Figure 4D*), nor a difference between closed and open loop locomotion onsets (*Figure 4—figure supplement 5*). Thus, it should be kept in mind that with any virus-based protein expression, the population of labeled cells is likely not random genetically, and that descriptor of 'layer 2/3 (or 5) neuron' should be understood to mean, a 'layer 2/3 (or 5) neuron that exhibits high expression levels under the artificial Ef1α promoter when used in an AAV delivery vector'.

Given our results, what are the implications for the computational role of acetylcholine in visual cortex? There are likely two aspects to this, the acute effect on cortical activity and the effect on plasticity. We find that acetylcholine was released in visual cortex as a locomotion state signal. This state signal could be used to gate visuomotor plasticity. In systems engaged in sensorimotor learning, plasticity related to feedback prediction errors should occur primarily during self-motion and not when passively experiencing sensory input. Visuomotor development is critically dependent on coupling of locomotion and sensory feedback (*Attinger et al., 2017*; *Held and Hein, 1963*; *Kaneko and Stryker, 2014*), and depends on local plasticity mechanisms in visual cortex (*Widmer et al., 2022*). Acetylcholine is known to gate plasticity in visual cortex (*Bear and Singer, 1986*; *Greuel et al., 1988*; *Gu and Singer, 1993*; *Kirkwood et al., 1999*) and could function to gate visuomotor plasticity to phases of self-motion. Given that acetylcholine release is associated with other types of movement as well (*Eggermann et al., 2014*; *Harrison et al., 2016*; *Lohani et al., 2022*; *Nelson and Mooney, 2016*; *Zhu et al., 2023*), this is consistent with acetylcholine functioning more generally as a movement state signal. We would predict that in visual cortex, this would primarily relate to movements that result in changes to visual input.

The absence of an acute effect of optogenetically stimulating cholinergic axons locally on the activity of visual cortex neurons is consistent with the idea that locomotion-driven neuronal activity in visual cortex (*Keller et al., 2012*; *Saleem et al., 2013*) is likely not driven by acetylcholine, but is the consequence of a combination of motor-related top-down input (*Leinweber et al., 2017*) and noradrenaline (*Polack et al., 2013*), another neuromodulator that is known to accompany locomotion onset (*Jordan and Keller, 2023*; *Reimer et al., 2016*). Consistent with previous reports (*Goard and Dan, 2009*), we find that acetylcholine acutely increases visuomotor responses more strongly in layer 5 neurons than in layer 2/3 neurons and that it decorrelates neuronal activity. One of the key questions here is why these effects are induced during locomotion. There are a number of explanations that have been proposed previously that often focus on increasing sensory gain to counteract the noisier visual input during locomotion (*Minces et al., 2017*; *Pinto et al., 2013*). We propose a slightly more specific explanation. This is based on the idea that the observed effects of acetylcholine on layer 5 can be explained by assuming the layer 5 network exhibits attractor dynamics and that acetylcholine modifies these attractor dynamics (*Kanamaru and Aihara, 2019*). The observed differences in the effect of acetylcholine on layer 5, where response strength increased, and layer 2/3, where they did not, are consistent with the interpretation that attractor dynamics are stronger in layer 5 than in layer 2/3. This would, also agree with a proposed microcircuit for predictive processing (*Keller and Mrsic-Flogel, 2018*), where layer 5 is postulated to maintain an internal representation (*Heindorf and Keller, 2022*). Thus, acetylcholine could act to increase the sensitivity and speed of response of the internal representation neurons to inputs arriving from outside the local network at the expense of reducing the influence among neurons in the local network. Such an acetylcholine-driven increase in the speed of transition between different internal representations would also be consistent with the finding that layer 5 neurons exhibit faster responses to the transitions between different stimuli (*Figure 7*). Assuming the internal representation in cortex is a substrate of conscious perception, as proposed by predictive processing, this would be consistent with the finding that anesthesia decouples layer 5 neurons from their inputs onto apical dendrites by cholinergic mechanisms (*Suzuki and Larkum, 2020*).

What makes this hypothesis all the more tantalizing is the observation that a decrease in cortical cholinergic activation is correlated with the severity of dementia in Alzheimer's disease (*Perry et al., 1978*). All three drugs currently approved for the management of dementia in Alzheimer's disease are cholinesterase inhibitors, drugs that increase acetylcholine levels. If cholinergic activation alters network dynamics to make the putative layer 5 attractor network more sensitive to external inputs – and hence more easily influenced by long-range cortical inputs – this could explain why cholinergic activation enhances memory retrieval. A decrease in cholinergic tone in Alzheimer's disease is also

consistent with the increases of correlation of neuronal activity in cortex observed in mouse models of the disease (*Korzhova et al., 2021*). Thus the cholinergic gating of the sensitivity of layer 5 to external inputs could constitute a possible mechanism for the cholinergic hypothesis for Alzheimer's disease (*Davies and Maloney, 1976*; *Francis et al., 1999*; *Hampel et al., 2019*; *Terry and Buccafusco, 2003*).

In summary, we show that cholinergic activity in visual cortex conveys a locomotion state signal, that preferentially augments the response of layer 5 neurons to external inputs and decorrelates activity across neurons, possibly to both sculpt the receptive field tuning of neurons to sensory stimuli as well as to regulate the influence of external input on internal representation to affect our conscious percept.

# Methods

**Key resources table**

| Reagent type (species) or resource | Designation | Source or reference | Identifiers | Additional information |
|---|---|---|---|---|
| Strain, strain background (adeno-associated virus) | AAV2/5-hSyn1-FLEx-axon-GCaMP6s ($10^{13}$–$10^{15}$ GC/ml) | FMI vector core | vector.fmi.ch | |
| Strain, strain background (adeno-associated virus) | AAV PHP.eB-Ef1α-DIO-GCaMP6s ($10^{11}$ GC/ml) | FMI vector core | vector.fmi.ch | |
| Strain, strain background (adeno-associated virus) | AAV2/1-hSyn-DIO-ChrimsonR-tdTomato ($10^{11}$–$10^{13}$ GC/ml) | FMI vector core | vector.fmi.ch | |
| Strain, strain background (adeno-associated virus) | AAV2/1-Ef1α-GCaMP6f-WPRE ($10^{11}$–$10^{14}$ GC/ml) | FMI vector core | vector.fmi.ch | |
| Strain, strain background (adeno-associated virus) | AAV2/1-Ef1α-DIO-hM3D(Gq)-mCherry ($10^{11}$ GC/ml) | FMI vector core | vector.fmi.ch | |
| Strain, strain background (adeno-associated virus) | AAV2/1-Ef1α-DIO-hM4D(Gi)-mCherry ($10^{11}$ GC/ml) | FMI vector core | vector.fmi.ch | |
| Strain, strain background (adeno-associated virus) | AAV2/9-hSyn-GRAB-ACh3.0 ($10^{13}$ GC/ml) | FMI vector core | vector.fmi.ch | |
| Strain, strain background (adeno-associated virus) | AAV2/1-Ef1α-DIO-tdTomato-WPRE ($10^{14}$ GC/ml) | FMI vector core | vector.fmi.ch | |
| Strain, strain background (adeno-associated virus) | AAV2/1-Ef1α-GFP-WPRE ($10^{12}$ GC/ml) | FMI vector core | vector.fmi.ch | |
| Chemical compound, drug | Clozapine-N-oxide (CNO) | Tocris | Cat# 4936–10 mg | DREADD activator |
| Chemical compound, drug | Fentanyl citrate | Actavis | CAS 990-73-8 | Anesthetic compound |
| Chemical compound, drug | Midazolam (Dormicum) | Roche | CAS 59467-96-8 | Anesthetic compound |
| Chemical compound, drug | Medetomidine (Domitor) | Orion Pharma | CAS 86347-14-0 | Anesthetic compound |
| Chemical compound, drug | Ropivacaine | Presenius Kabi | CAS 132112-35-7 | Analgesic compound |
| Chemical compound, drug | Lidocaine | Bichsel | CAS 137-58-6 | Analgesic compound |
| Chemical compound, drug | Buprenorphine | Reckitt Benckiser Healthcare | CAS 52485-79-7 | Analgesic compound |

*Continued on next page*

*Continued*

| Reagent type (species) or resource | Designation | Source or reference | Identifiers | Additional information |
|---|---|---|---|---|
| Chemical compound, drug | Humigel | Virbac | - | Ophthalmic gel |
| Chemical compound, drug | Flumazenil (Anexate) | Roche | CAS 78755-81-4 | Anesthetic antagonist |
| Chemical compound, drug | Atipamezole (Antisedan) | Orion Pharma | CAS 104054-27-5 | Anesthetic antagonist |
| Chemical compound, drug | Metacam | Boehringer Ingelheim | CAS 71125-39-8 | Analgesic compound |
| Chemical compound, drug | N-Butyl-2-cyanoacrylate | Braun | CAS 6606-65-1 | Histoacryl |
| Chemical compound, drug | Dental cement (Paladur) | Heraeus Kulzer | CAS 9066-86-8 | |
| Genetic reagent *Mus musculus* | *C57BL/6* | Charles River | - | |
| Genetic reagent *Mus musculus* | *B6J.129S6-Chat^{tm2(Cre)Lowl}/MwarJ* Alias used here: ChAT-IRES-Cre | Jackson Laboratories | RRID:IMSR_JAX:028861 | Cre expression in cholinergic neurons |
| Software, algorithm | MATLAB (2020b) | The MathWorks | RRID:SCR_001622 | Data analysis |
| Software, algorithm | LabVIEW | National Instruments | RRID:SCR_014325 | Hardware control |
| Software, algorithm | Two-photon acquisition software | Keller laboratory | https://sourceforge.net/projects/iris-scanning/ | Data acquisition |
| Software, algorithm | Image data processing software | Keller laboratory | https://sourceforge.net/p/iris-scanning/calliope/HEAD/tree/ | Data processing |
| Software, algorithm | Python | https://www.python.org/ | RRID:SCR_008394 | Virtual reality |
| Software, algorithm | Panda3D | https://www.panda3d.org/ | N/A | Virtual reality |
| Other | Virtual reality and two-photon setup | *Leinweber et al., 2014*; *Leinweber et al., 2017* | DOI: https://doi.org/10.3791/50885, https://doi.org/10.1016/j.neuron.2017.08.036 | Hardware setup |
| Other | OBIS 673 nm LX | Coherent | Cat#1187194 | Optogenetic stimulation laser |
| Other | LED | Prizmatix | UHP-T-595 | Sham stimulation |
| Other | Titanium headplate | FMI/ETHZ workshop | N/A | Mice head-fixation |
| Other | Dental drill | Meisinger | N/A | For craniotomy |

## Mice

All mice used in experiments described in the paper were ChAT-IRES-Cre (*Rossi et al., 2011*) heterozygotes, kept on a C57BL/6 background. A total of 64 mice, both male and female, 6–16 weeks old at the start of the experiment, were used. See *Supplementary file 2* for details of mouse inclusion for the different figures. Between experiments, mice were group-housed in a vivarium (light/dark cycle: 12/12 hr). All animal procedures were approved by and carried out in accordance with the guidelines laid by the Veterinary Department of the Canton of Basel-Stadt, Switzerland.

## Surgery

For all surgical procedures, mice were anesthetized with a mixture of fentanyl (0.05 mg/kg; Actavis), midazolam (5.0 mg/kg; Dormicum, Roche), and medetomidine (0.5 mg/kg; Domitor, Orion) injected intraperitoneally. Analgesics were applied perioperatively (2% lidocaine gel, meloxicam 5 mg/kg) and postoperatively (buprenorphine 0.1 mg/kg, meloxicam 5 mg/kg). Eyes were covered with ophthalmic gel (Virbac Schweiz AG). Cranial windows were implanted as previously described (*Keller et al., 2012*;

*Leinweber et al., 2014*). Briefly, using a dental drill, a 4 mm craniotomy was made over the right visual cortex, centered 2.5 mm lateral and 0.5 mm anterior to lambda. After injection of AAV vectors, the exposed cortex was sealed with a 4 mm circular glass coverslip and glued in place using gel super-glue (Ultra Gel, Pattex). The remaining exposed surface of the skull was covered with Histoacryl (B. Braun), and a titanium head bar was fixed to the skull using dental cement (Paladur, Heraeus Kulzer). After surgery, anesthesia was antagonized by a mixture of flumazenil (0.5 mg/kg; Anexate, Roche) and atipamezole (2.5 mg/kg; Antisedan, Orion Pharma) injected intraperitoneally.

### Axonal labeling

To image the activity of basal forebrain cholinergic axons in visual cortex, we expressed a calcium indicator in basal forebrain cholinergic neurons. Surgery was performed as described above, and either an AAV2/5-hSyn1-FLEx-axon-GCaMP6s ($10^{13}$ GC/ml) or an AAV-PHP.eB-Ef1α-DIO-GCaMP6s ($10^{12}$ GC/ml) was injected at four locations ipsilateral to the imaging site (AP, ML, DV (in mm): +1.1, +0.1, –3.7 (medial septum); +1.1, +0.1, –4.1 and 0.8, +0.1, –4.2 (vertical limb of the diagonal band); +0.6, +0.6, –4.9 (horizontal limb of the diagonal band)) in ChAT-IRES-Cre mice.

### GRAB imaging

We injected an AAV vector carrying GRAB-ACh 3.0, AAV2/9-hSyn-GRAB-ACh3.0 ($10^{13}$ GC/ml), in visual cortex. A 4 mm craniotomy was made over the right visual cortex and sealed with a glass coverslip as described above.

### Chemogenetic experiments

To manipulate the activity of the basal forebrain cholinergic system, we used chemogenetic DREADD inhibitor AAV2/1-EF1α-DIO-hM4D(Gi)-mCherry ($10^{11}$ GC/ml) or DREADD activator AAV2/1-EF1α-DIO-hM3D(Gq)-mCherry ($10^{11}$ GC/ml). AAVs were injected in the anterior basal forebrain with coordinates as used for axonal imaging described above. DREADDs were stimulated using Clozapine N-oxide (CNO), which was dissolved in DMSO, diluted with saline, and injected intraperitoneally at a dose of 0.5 mg/kg body weight. Activity from each imaging site in the DREADD series was acquired starting 30-60 min after CNO injection.

### Optogenetic experiments

We injected AAV2/1-hSyn-DIO-ChrimsonR-tdTomato ($10^{11}$ GC/ml) into basal forebrain at the same coordinates as for axonal imaging described above, and injected AAV2/1-Ef1α-GCaMP6f-WPRE ($10^{11-14}$ GC/ml) into visual cortex for imaging neurons in layer 2/3 and layer 5. In no-opsin control mice, instead of ChrimsonR, we injected an AAV2/1-DIO-tdTomato-WPRE ($10^{14}$ GC/ml) virus. ChrimsonR stimulation and functional imaging of GCaMP6f-expressing neurons was done as previously described (*Attinger et al., 2017*). For imaging, we used a modified Thorlabs B-Scope with a 12 kHz resonance scanner (Cambridge Technology). The illumination source for the optogenetic stimulation was a 637 nm laser (OBIS LX, Coherent). We used a dichroic mirror (ZT775sp-2p, Chroma) to combine the two-photon laser and stimulation laser. A second long-pass dichroic mirror (F38-555SG, Semrock) was used to split the GCaMP emission from both illumination light sources. Light leak from the 637 nm stimulation laser was reduced by synchronizing the stimulation laser to the turnaround times of the resonant scanner (during which imaging data were not acquired). Lastly, amplified PMT signals were digitally bandpass filtered at 80 MHz to reduce the effect of ringing in the amplifier. This allowed for near stimulation-artifact free synchronous imaging and optogenetic stimulation. For all experiments with stimulation of cholinergic axons locally in visual cortex, we used a square-wave pulse of 1 s duration and 10 mW/mm² power (measured at objective). The median inter-trial interval for different stimulation events was 15 s for optogenetic stimulation-only trials, 12 s for optogenetic stimulation with grating onset, 35 s for optogenetic stimulation with visuomotor mismatch, and 45 s for optogenetic stimulation with locomotion onset in the closed loop. To measure correlation changes, we employed a longer laser stimulation window of 1 min, at 40 Hz (50% duty cycle), with an average power output of 20 mW/mm² (measured at objective). For sham stimulation, we used an optical guide coupled to a high-power LED (UHP-T-595, Prizmatix) to diffusely illuminate the mouse and surrounding virtual reality setup.

## Virtual reality environment

The virtual reality setup is based on the design of Dombeck and colleagues (*Dombeck et al., 2007*). Briefly, mice were head-fixed and free to run on an air-supported spherical treadmill. The rotation of the ball was restricted around the vertical axis with a pin. The virtual reality environment was projected onto a toroidal screen covering approximately 240° horizontally and 100° vertically of the mouse's visual field, using a projector (Samsung SP-F10M) synchronized to the resonant scanner of the two-photon microscope. The virtual environment consisted of an infinite corridor with walls patterned with vertical sinusoidal gratings with a spatial frequency of approximately 0.04 cycles per degree (*Leinweber et al., 2014*). In closed loop condition, the locomotion of the mouse was coupled to movement along a virtual tunnel. In open loop condition, we uncoupled the two and replayed the visual flow from a preceding closed loop condition. In grating condition, we presented full field drifting gratings (0°, 45°, 90°, 270°, moving in either direction) in a pseudo-random sequence. Grating stimuli were presented for between 2 s and 3 s. In the inter-stimulus interval (between 2 s and 4 s), mice were shown a gray screen with average luminance matched to that of the grating stimuli.

## Eye tracking and facial movement measurement

During all experiments, we recorded the mouse's left eye (contralateral to the imaged hemisphere) with a CMOS infrared camera at 30 Hz frame rate. The pupil was backlit by the 930 nm laser used for two-photon imaging. We calculated pupil diameter offline by fitting a circle to the pupil. Frames with occluded pupil were excluded from the analysis. To extract facial movements, we computed the average image displacement in different image patches on the face of the mouse.

## Two-photon microscopy

Functional two-photon calcium imaging was performed using custom-built two-photon microscopes (*Leinweber et al., 2014*). The illumination source was a tunable femtosecond laser (Insight, Spectra-Physics or Chameleon, Coherent) tuned to 930 nm. Emission light was band-pass filtered using a 525/50 filter for GCaMP and a 607/70 filter for tdTomato/mCherry (Semrock) and detected using a GaAsP photomultiplier (H7422, Hamamatsu). Photomultiplier signals were amplified (DHPCA-100, Femto), digitized (NI5772, National Instruments) at 800 MHz, and band-pass filtered at 80 MHz using a digital Fourier-transform filter implemented in custom-written software (see **Key resources table**) on an FPGA (NI5772, National Instruments). The scanning system of the microscopes was based either on a 12 kHz or an 8 kHz resonant scanner (Cambridge Technology). Images were acquired at a resolution of 750×400 pixels (60 Hz or 40 Hz frame rate, respectively), and a piezo-electric linear actuator (P-726, Physik Instrumente) was used to move the objective (Nikon 16 x, 0.8 NA) in steps of 15 μm between frames to acquire images at 4 different depths. This resulted in an effective frame rate of 15 Hz or 10 Hz, respectively. The field of view was 375 μm×300 μm.

## Extraction of neuronal activity

Calcium imaging data were processed as previously described (*Keller et al., 2012*) and all data analysis was done in MATLAB (MathWorks). Briefly, raw images were full-frame registered to correct for lateral brain motion. Neurons and axons were manually selected based on mean and maximum fluorescence images. For GRAB imaging, neuropil and blood vessels were manually selected. For a subset of mice, the apical dendrites of layer 5 neurons were marked in layer 2/3 by manually tracing them to their soma in layer 5 using a z-stack acquired at the end of an experiment. Raw fluorescence traces were corrected for slow drift in fluorescence using an 8$^{th}$-percentile filtering with a 66 s (or 1000 frames) window (*Dombeck et al., 2007*). ΔF/F traces were calculated as mean fluorescence in a selected region of every imaging frame, subtracted, and normalized by the overall median fluorescence. All neuronal calcium activity data was acquired at 15 Hz. For axonal imaging, 9 (of 25 sites) were imaged at 10 Hz, and analytically resampled at 15 Hz using a polyphase antialiasing filter to make the time-base compatible with the rest of the axonal imaging data.

## Data analysis

All data analysis was done using custom scripts written in MATLAB (MathWorks). To quantify the average population response traces, we first calculated the average event-triggered fluorescence trace for each region of interest (ROI). The responses of all ROIs was then averaged and baseline-subtracted.

Locomotion onset was defined as locomotion velocity crossing a threshold of 0.25 cm/s for at least 1 s, while having been below the threshold for 1 s before. The same criteria were used to define visual flow onsets in the open loop condition using visual flow speed. Visuomotor mismatch responses were probed by presenting brief 1 s full field visual flow halts in the closed loop condition. For a mismatch event to be included in the analysis, mice had to be locomoting uninterrupted above threshold (0.25 cm/s) from –0.5 s to +1 s after the event onset. Additionally, for a ROI to be included for analysis of the response to a particular event, it had to have at least 5 onsets to the event.

In *Figure 1E*, the calcium responses were baseline subtracted using a –1 s to –0.5 s window relative to locomotion onset and sorted based on the average response in the +0.5 s to +1.5 s activity window. In *Figure 1F–H*, the activity was averaged over all axons, with a baseline subtraction window of –1 s to –0.5 s relative to onset. In *Figure 1I*, the baseline window was –0.5 s to 0 s relative to mismatch onset (to include the locomotion-related response in the baseline), and a 95% confidence interval was calculated from randomly drawn timepoints during locomotion. In *Figure 1J*, the locomotion onset response is plotted across different visuomotor conditions, and the baseline window was –1 s to –0.5 s. In *Figure 1K*, cholinergic axons ware classified as responsive to a particular stimulus if the mean activity in a window +0.5 s to +1.5 s after stimulus onset was significantly different (paired two-sample t-test over onsets, $p<0.05$) from the mean of the baseline window (–1 s to –0.5 s).

Locomotion offset was defined as the threshold crossing at which locomotion velocity decreased below 0.25 cm/s for at least 1 s after mice ran above the threshold for at least 1 s. To isolate locomotion bouts, we identified consecutive locomotion onset and offset events that were separated by at least 4 s. In *Figure 2C*, we computed correlations in average calcium activity (or locomotion velocity) between locomotion onset and offset across locomotion bouts. Onset responses were defined as the average population activity (or locomotion velocity) +0.5 s to +1.5 s after locomotion onset with a baseline of –1 s to –0.5 s. Offset responses were similarly defined as the average population activity (or locomotion velocity) over –1.5 s to –0.5 s before locomotion offset with the same baseline as for bout onset. In *Figure 2D*, axons were sorted based on their average activity at locomotion onset (defined as above), and the same sort order was preserved for plotting the average activity during the bout and at bout offset. In *Figure 2E*, locomotion velocity was binned into 100 evenly spaced bins in the range of 0 cm/s to 6.25 cm/s, and the activity of cholinergic axons in each bin was averaged. In *Figure 2F*, the correlation of the activity of each cholinergic axon was computed against the mouse locomotion velocity, and this was compared against the correlation computed with a binarized version of locomotion velocity, binarized using a threshold of 0.25 cm/s.

In *Figures 3B, C and 4B* with pupil data, we excluded frames with blinks. For variability and correlation analysis involving pupil diameter, sites with less than 30 s of pupil data in a visuomotor condition were excluded.

In *Figure 4A–H*, to probe for the influence of cholinergic stimulation on responses in visual cortex, the optogenetic stimulation laser was turned on for 1 s coincident with the onset of mismatch or grating stimuli. Further, to isolate the influence of cholinergic stimulation based on the locomotion state, an instance of grating stimulus presentation was defined as occurring during 'locomotion' if mice had a locomotion velocity that was above a threshold (0.25 cm/s) in the time-window of –0.5 s to +1 s from the stimulus onset, otherwise it was defined as occurring while the mouse was 'stationary'. Note, the same criteria to define locomotion state was used to isolate visuomotor mismatches, and also during control optogenetic stimulation experiments (*Figure 4—figure supplement 3A–E*). For stimulation on locomotion onset, we detected locomotion onsets based on locomotion velocity in real-time. The average delay between locomotion onsets and laser stimulation onsets was 406 ms ± 41 ms (mean ± SEM). For *Figure 4—figure supplement 2*, we selected the 10% most responsive neurons to gratings, visuomotor mismatch, and locomotion onset. This selection was based on their mean response over half of the trials without optogenetic stimulation (response window: +0.5 s to +2.0 s; baseline window: –0.5 s to 0 s from stimulus onset). We then compared their response over the remaining trials to those with optogenetic stimulation.

In *Figure 4—figure supplement 4*, for each neuron, we determined the preferred orientation using half the trials while the mice were stationary, and plotted responses relative to this preferred orientation in the other half of the stationary trials, or while the mice were locomoting, or during optogenetic stimulation of cholinergic axons while stationary. As before, a velocity threshold of 0.25 cm/s was used to separate stationary trials from locomoting.

In *Figure 6A–F*, to compute the average pairwise correlation in the activity of neurons, we first split the activity trace based on the locomotion state of the mouse and the presence of optogenetic stimulation. Then, for each condition, we computed the average over all pairwise correlations of a given neuron with every other neuron from the same imaging site. For the chemogenetic experiment, we further compared the change in average pairwise correlation before and after injection of the DREADD ligand. We only included those sites in this analysis which had at least 15 s behavior for each of the two compared conditions.

We defined latency to activation on locomotion onset (*Figure 1—figure supplement 1*) for a neuron or axon as the time after onset when the calcium activity, averaged across onsets, first was 2 standard deviations above baseline, and remained above this threshold thereafter for at least 1 s. Neurons that did not reach the threshold within the analysis window –2 s to +3 s and were not significantly responsive to locomotion onset (paired two-sample t-test over onsets, $p < 0.05$, between baseline –1 s to –0.5 s and response +0.5 s to +1.5 s), were not included in the analysis.

Response latency to grating stimuli (*Figure 7*) was defined for a neuron as the time after onset when the calcium activity, averaged across onsets, first was 2 standard deviations above baseline, and remained above this threshold for at least 1/3 s. Neurons that did not reach the threshold within the analysis window of 0 s to +2 s (equal to the duration of the grating presentation) were not included in the analysis.

## Statistical analysis

All statistical information for the tests performed in the manuscript is provided in *Supplementary file 1*. Unless stated otherwise, the shading indicates the standard error of the mean across axons or neurons. For analysis where the experimental unit was an axon, neuron, or pair of neurons, we used hierarchical bootstrap (*Saravanan et al., 2020*) for statistical testing due to the nested nature (axons/neurons and mice) of the data. Briefly, we first resampled the data (with replacement) at the level of imaging sites, and then, from the selected sites, resampled for axons (or neurons or neuron pairs). We then computed the mean of this bootstrap sample and repeated this 10 000 times (or 1000 times when comparing response traces in *Figure 4*, *Figure 4—figure supplement 2* and *Figure 4—figure supplement 3*) to generate a bootstrap distribution of the mean estimate. For paired tests, the p-value was defined as the proportion of bootstrap samples higher (or lower, depending on the hypothesis) than zero. For unpaired tests, the distribution of the mean for the two variables were compared as the proportion of values higher (or lower, depending on the hypothesis). For analysis where the experimental unit was the imaging site, we first tested for normality using the Kolmogorov-Smirnov test at a significance level of 5%. For datasets that did not pass the test for normality, the medians were compared with the Wilcoxon rank-sum test or signed rank test, as applicable. Otherwise, the means were compared using paired or unpaired t-tests, as applicable.

## Acknowledgements

We thank all the members of the Keller lab for their discussion and support. This project has received funding from the Swiss National Science Foundation (GBK), the Novartis Research Foundation (GBK), and the European Research Council (ERC) under the European Union's Horizon 2020 research and innovation program (grant agreement No 865617) (GBK).

## Additional information

### Funding

| Funder | Grant reference number | Author |
| --- | --- | --- |
| European Research Council | 865617 | Georg B Keller |
| Novartis Stiftung für Medizinisch-Biologische Forschung | | Georg B Keller |

| Funder | Grant reference number | Author |
|---|---|---|
| Swiss National Science Foundation | | Georg B Keller |

The funders had no role in study design, data collection and interpretation, or the decision to submit the work for publication.

## Author contributions

Baba Yogesh, Conceptualization, Resources, Data curation, Software, Formal analysis, Validation, Investigation, Visualization, Methodology, Writing - original draft, Project administration, Writing - review and editing; Georg B Keller, Resources, Supervision, Funding acquisition, Writing - original draft, Writing - review and editing

## Author ORCIDs

Baba Yogesh (i) http://orcid.org/0000-0003-1224-6035
Georg B Keller (i) https://orcid.org/0000-0002-1401-0117

## Ethics

All animal procedures were approved by and carried out in accordance with the guidelines laid by the Veterinary Department of the Canton of Basel-Stadt, Switzerland. License 2573.

Reviewer #1 (Public review): https://doi.org/10.7554/eLife.89986.5.sa1
Reviewer #2 (Public review): https://doi.org/10.7554/eLife.89986.5.sa2
Author response https://doi.org/10.7554/eLife.89986.5.sa3

# Additional files

## Supplementary files

• Supplementary file 1. Statistical information on all analysis. For statistical comparisons we used paired or unpaired t-tests, or hierarchical bootstraps as indicated in the table below. The units (ROIs, sites, or mice) over which testing was done are boldfaced. For hierarchical bootstraps the two levels of units used are boldfaced.
• Supplementary file 2. List of mice used in the different experimental groups.
• MDAR checklist

## Data availability

All data generated and analysed during this study have been uploaded to Zenodo with the URL https://zenodo.org/records/12632129.

The following dataset was generated:

| Author(s) | Year | Dataset title | Dataset URL | Database and Identifier |
|---|---|---|---|---|
| Yogesh B, Keller GB | 2024 | Cholinergic input to mouse visual cortex signals a movement state and acutely enhances layer 5 responsiveness | https://doi.org/10.5281/zenodo.12632129 | Zenodo, 10.5281/zenodo.12632129 |

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
