## [Editor Report · eLife assessment]

This **fundamental** study by Yogesh and Keller provides a set of results describing the response properties of cholinergic input and its functional impacts in the mouse visual cortex. They found that cholinergic inputs are elevated by locomotion in a binary manner regardless of locomotor speeds, and activation of cholinergic input differently modulated the activity of Later 2/3 and Layer 5 visual cortex neurons induced by bottom-up (visual stimuli) and top-down (visuomotor mismatch) inputs. The experiments are cutting-edge and well-executed, and the results are **convincing**.

---

## [Referee Report · Reviewer #1 (Public review)]

The paper submitted by Yogesh and Keller explores the role of cholinergic input from the basal forebrain (BF) in the mouse primary visual cortex (V1). The study aims to understand the signals conveyed by BF cholinergic axons in the visual cortex, their impact on neurons in different cortical layers, and their computational significance in cortical visual processing. The authors employed two-photon calcium imaging to directly monitor cholinergic input from BF axons expressing GCaMP6 in mice running through a virtual corridor, revealing a strong correlation between BF axonal activity and locomotion. This persistent activation during locomotion suggests that BF input provides a binary locomotion state signal. To elucidate the impact of cholinergic input on cortical activity, the authors conducted optogenetic and chemogenetic manipulations, with a specific focus on L2/3 and L5 neurons. They found that cholinergic input modulates the responses of L5 neurons to visual stimuli and visuomotor mismatch, while not significantly affecting L2/3 neurons. Moreover, the study demonstrates that BF cholinergic input leads to decorrelation in the activity patterns of L2/3 and L5 neurons.

This topic has garnered significant attention in the field, drawing the interest of many researchers actively investigating the role of BF cholinergic input in cortical activity and sensory processing. The experiments and analyses were thoughtfully designed and conducted with rigorous standards, providing evidence of layer-specific differences in the impact of cholinergic input on neuronal responses to bottom-up (visual stimuli) and top-down inputs (visuomotor mismatch).

---

## [Referee Report · Reviewer #2 (Public review)]

The manuscript investigates the function of basal forebrain cholinergic axons in mouse primary visual cortex (V1) during locomotion using two-photon calcium imaging in head-fixed mice. Cholinergic modulation has previously been proposed to mediate the effects of locomotion on V1 responses. The manuscript concludes that the activity of basal forebrain cholinergic axons in visual cortex provides a signal which is more correlated with binary locomotion state than locomotion velocity of the animal and finds no evidence for modulation of cholinergic axons by locomotion velocity. Cholinergic axons did not seem to respond to grating stimuli or visuomotor prediction error. Optogenetic stimulation of these axons increased the amplitude of responses to visual stimuli and decreased the response latency of layer 5 excitatory neurons, but not layer 2/3 neurons. Moreover, optogenetic or chemogenetic stimulation of cholinergic inputs reduced pairwise correlation of neuronal responses. These results provide insight into the role of cholinergic modulation to visual cortex and demonstrate that it affects different layers of visual cortex in a distinct manner. The experiments are well executed and the data appear to be of high quality.

---

## [Author Response]

Author response:

The following is the authors’ response to the previous reviews.

**Reviewer #1 (Recommendations For The Authors):**
The author has addressed all the concerns I have raised.I have only one minor suggestion.We would argue both a gray screen and a grating are visual stimuli. ... We concur, our data only address one of many possible transitions, but it is a switch between distinct visual stimuli that is sped up by ACh.Thank you for clarifying this.Following my comment in the previous review, the author has revised the abstract as follows: (Before) "Our results suggest that acetylcholine augments the responsiveness of layer 5 neurons to inputs from outside of the local network, enabling faster switching between internal representations during locomotion."(After) "Based on this we speculate that acetylcholine augments the responsiveness of layer 5 neurons to inputs from outside of the local network, possibly enabling faster switching between internal representations during locomotion."My previous comment concerned specifically the latter part, "enabling faster switching between internal representations during locomotion", and, in fact, their data fully support the first part, "acetylcholine augments the responsiveness of layer 5 neurons to inputs from outside of the local network". Thus, I suggest the following sentence:"Our results suggest that acetylcholine augments the responsiveness of layer 5 neurons to inputs from outside of the local network, possibly enabling faster switching between internal representations during locomotion."

Thank you for clarifying. We have changed as suggested.

**Reviewer #2 (Recommendations For The Authors):**
I thank the authors for the clarification regarding the distribution of running speeds in the study. I do agree that 30 cm/s is indeed fast for head-fixed locomotion. My concern is that while all mice contribute to the low locomotion velocity bin, the high locomotion velocity bin is dominated by a subset of animals, since not all mice reached high locomotion speeds. Therefore, the comparison between low, intermediate and high locomotion velocities includes data from different cohorts of animals and variability across animals may confound the analysis of cholinergic axon activity. However, the manuscript is carefully worded to emphasize lack of evidence (e.g. "we found no evidence of an increase in calcium activity between low and high locomotion velocities") and I have revised my summary in the public review to reflect this.I thank the authors for including the scatterplots of single neuron responses locomotion and optogenetic stimulation, which illustrate their heterogeneity. I am surprised that the axes are limited to 20% deltaF/F as visual responses recorded using GCaMP6f often exceed 100% deltaF/F .

There are definitely neurons with responses larger than 20% dF/F0, but it is a small fraction. There are two considerations relevant to assessing dF/F amplitudes. First, in our hands trial averaged dF/F0 responses tend to be below 30% even for the most responsive neurons (trial averaging convolves response amplitude and response reliability). The reviewer is probably thinking of single trial responses often shown as raw data that can exceed 100s of %. Second, different published variants for calculating dF/F0 can result in a spectrum of values that varies by up to a factor of 10. This is largely a consequence of the choice of F0 and preprocessing related to correcting slow drifts in signal strength (originally motivated by photobleaching). Attempting to compare dF/F0 across labs is unfortunately a futile effort in absence of standardized way of calculating it.

Allow me to clarify how evaluating the effects of optogenetic stimulation and locomotion without analyzing them at the level of individual neurons could result in misleading conclusions. I will use the effects of cholinergic responses on grating responses as an example but this concern applies equally to the other analyses. The manuscript reports that "in layer 2/3, optogenetic activation of cholinergic axons did not result in a detectable increase in grating onset responses (Figure 4C), while the responses of layer 5 neurons to the same stimulus increased with concurrent optogenetic activation of cholinergic axons." As the Figure R2C-D illustrates, only a minority of L2/3 neurons are excited by the grating in baseline conditions, while the vast majority are either suppressed or non-responsive. This is expected, as it is well established that visual responses in layer 2/3 are sparse. If responses of the small subset of L2/3 neurons that are activated by the grating were enhanced, it may not be apparent in the population average presented in the manuscript. In contrast, since a larger fraction of L5 neurons is excited by the grating, enhancement of grating responses may be easier to detect. In other words, the effects of optogenetic stimulation may be to boost the responses of those neurons that are activated by the grating and the difference between L2/3 and L5 lies simply in the proportion of activated neurons. I do not mean to argue in favour of this specific scenario but simply present it so as to illustrate the way in which considering population averages alone may be misleading.While the authors state in their response that "all relevant and clear conclusions are already captured by the mean differences shown in Figure 4", the evidence supporting this statement is not presented in the manuscript. Most importantly, it is essential to determine whether the neurons that show significant activation in response to gratings (Figure 4C-D), mismatch (Figure 4E-F) or locomotion (Figure 4G-H), are affected by optogenetic stimulation in the same way as the population average.

We have added the analysis suggested as Figure S6. Consistent with the population averages, even within the subset of layer 2/3 neurons most responsive to specific inputs, we found no detectable increase in responsiveness upon optogenetic stimulation of cholinergic axons.